# Residue-resolved monitoring of protein hyperpolarization at sub-second time resolution

Mattia Negroni[1] & Dennis Kurzbach ![ORCID] [1]✉

Signal-enhancement techniques for NMR spectroscopy are important to amplify the weak resonances provided by nuclear spins. Recently, 'hyperpolarization' techniques have been intensively investigated. These provide nuclear spin states far from equilibrium yielding strong signal boosts up to four orders of magnitude. Here we propose a method for real-time NMR of 'hyperpolarized' proteins at residue resolution. The approach is based on dissolution dynamic nuclear polarization (*d*-DNP), which enables the use of hyperpolarized buffers that selectively boost NMR signals of solvent-exposed protein residues. The resulting spectral sparseness and signal enhancements enable recording of residue-resolved spectra at a 2 Hz sampling rate. Thus, we monitor the hyperpolarization level of different protein residues simultaneously under near-physiological conditions. We aim to address two points: 1) NMR experiments are often performed under conditions that increase sensitivity but are physiologically irrelevant; 2) long signal accumulation impedes fast real-time monitoring. Both limitations are of fundamental relevance to ascertain pharmacological relevance and study protein kinetics.

[1] Faculty of Chemistry, Institute of Biological Chemistry, University Vienna, Währinger Str. 38, 1090 Vienna, Austria. ✉email: dennis.kurzbach@univie.ac.at

Processes involving protein interactions such as enzymatic reactions[1,2], biomineralization[3,4], or molecular recognitions[5,6] evolve in time toward an equilibrium state. However, it is notoriously challenging to access this temporal dimension while simultaneously maintaining high structural resolution[6–9]. This is not least because nuclear magnetic resonance (NMR) spectroscopy, the central method to access protein structures in solution, typically focuses on systems in chemical equilibrium[10,11]. This limitation is imposed by weak signal intensities that necessitate long signal averaging periods and thus impede time-resolved studies. Therefore, to adequately describe processes that evolve over time, methods are required that improve NMR signal intensities and, at the same time, capitalize on the method's intrinsically high (atomistic) resolution. We here propose such a method based on dissolution dynamic nuclear polarization (d-DNP)-boosted NMR and spin hyperpolarization[12–14].

Indeed, hyperpolarization, in particular dissolution d-DNP has recently been developed to enhance NMR signals in residue-resolved studies of proteins[15–20] and nucleic acids[21] at substantially improved signal intensities. To this methodological portfolio, we add a time aspect.

Due to the implementation of d-DNP as an 'ex-situ' technique, with an external hyperpolarization setup, it is applicable to almost any type of molecule in solution. It provides up to >10.000-fold enhanced signal intensities. As a result, this method enables access to time-resolved NMR data on milliseconds to seconds timescales while guaranteeing applicability to a broad spectrum of target molecules. Therefore, d-DNP has not yet found many applications in biomolecular NMR despite these exceptional benefits. This is not least since one has to give up the major advantage of solution-state protein NMR, namely residue-resolution. Indeed, the latter stimulated a large share of the timely NMR developments, including the development of a 1.2 GHz NMR spectrometer[22].

In this regard, d-DNP provides significantly improved signal intensities outdating the need for signal averaging. Spectra can be recorded rapidly, i.e., within milliseconds to seconds, yet only when relying on one-dimensional detection. Thus, acquiring NMR spectra in real-time NMR becomes possible, yet residue-resolved 2D or 3D spectra of proteins can typically not be recorded. Vice-versa, hyperpolarized spins have a limited lifetime (typically seconds to minutes), such that detection is possible only within a short time window. Hence, when recording fast 2D or 3D protein spectra by acquiring several signals to construct the indirectly detected dimensions, measuring a single spectrum takes too long for time-resolved measurement series.

Hence, herein, we present a strategy that aims to overcome this predicament by using hyperpolarized water to enhance NMR proton signals in proteins.

The use of hyperpolarized water to boost signal intensities in multidimensional NMR spectra of protein was first suggested by Frydman et al. in 2014[23]. In 2017, the approach was shown to be capable of yielding residue-resolved spectra of proteins and intrinsically disordered proteins[15,17], yet, only a single static spectrum could be detected. At the same time, it was shown by Hilty et al. that signal-boosted and time-resolved spectra could be obtained, albeit low-resolution[16]. Later, the hyperpolarized water technique was expanded to folded targets[24]. Since then, several applications of the technique have been developed, including the characterization of folding intermediates[25], exchange processes[26], and membrane interactions[18]. Herein, the use of hyperpolarized water to boost biomolecular NMR spectra is expanded by enabling both time- and residue-resolution in a single experiment. Similar approaches have also been successfully implemented for RNAs[21].

The development presented herein addresses two often encountered limitations of conventional solution-state NMR: (1)

Experiments are often performed under conditions that increase sensitivity but are physiologically not relevant (low pH, low temperature) and; (2) signal accumulation over long periods impedes the determination of fast (on the order of milliseconds to seconds) real-time monitoring. Both limitations are of equal fundamental relevance: interaction studies under non-native conditions are of limited pharmacological relevance, and the key to the function of proteins often resides in their interaction kinetics.

Owing to this signal boost achievable with hyperpolarized water, we were yet able to achieve simultaneously time- and residue-resolved NMR spectra of proteins under near-physiological conditions.

## Results and discussions

Our experimental strategy is based on two key concepts: hyperpolarized buffers and selective detection.

(1) *Hyperpolarized sample buffers*[27,28]. These can be used to enhance the signals in protein NMR. Hyperpolarization is here understood as a non-equilibrium state where the nuclear spins constructively align along the magnetic field $B_0$ to yield stronger signal amplitudes. This signal boost can be transferred from a hyperpolarized solvent to a protein of interest through chemical exchange and NOE effects[24,26]. With such techniques, substantial signal enhancements of several orders of magnitude can be achieved by slightly off-resonance microwave irradiation of the radical ESR line.

(2) *Selective detection*[17,24]. It is possible to select a small subset of hyperpolarized residues for detection using tailored NMR pulse sequences. As will be shown, spectra can be rendered so sparse that real-time monitoring at high sampling rates of individual amino acids becomes possible if boosted by hyperpolarization.

In the following, these two concepts are going to be outlined first. Then, we demonstrate how to combine them to achieve hyperpolarized real-time NMR of proteins.

(i) *Dissolution dynamic nuclear polarization as a tool to enhance protein signals*

The experimental strategy for signal amplification of proteins *via* hyperpolarized buffers has been laid out in a series of recent publications[15–18,24,25,29]. The method is based on the dissolution d-DNP technique. The procedure generally consists of three steps (Fig. 1):

(1) preparation of hyperpolarized, i.e., NMR-signal amplified water protons by DNP at cryogenic temperatures (here $T_{DNP} = 1.2$ K) and high magnetic field (here $B_{0,DNP} = 6.7$ T) achieved by off-resonance microwave irradiation (here $\nu_{DNP} = 188.38$ GHz).

(2) After the build-up, the hyperpolarized $H_2O$ is dissolved with a burst of superheated $D_2O$ and rapidly (here $t_{transfer} = 1.5$ s) injected into an NMR tube waiting in a high-field detection spectrometer (here $B_{0,NMR} = 18.8$ T, $T_{NMR} = 310$ K), where it is mixed in-situ with a solution of the target protein. After completion of the mixing process, NMR detection is triggered. Here we acquired time series of $^1H^N$-selective $^{15}N$-edited 1D $^1H$ spectra (similar to the first transient in a BEST-HMQC experiment)[30] for 50 s after mixing. The pulse sequence was applied without any water suppression in order to not accelerate the depletion of the hyperpolarized pool of protons[30].

(3) During the detection period, proton chemical and magnetic (i.e., nuclear Overhauser effects; NOE) exchange processes between the buffer and the target protein introduce

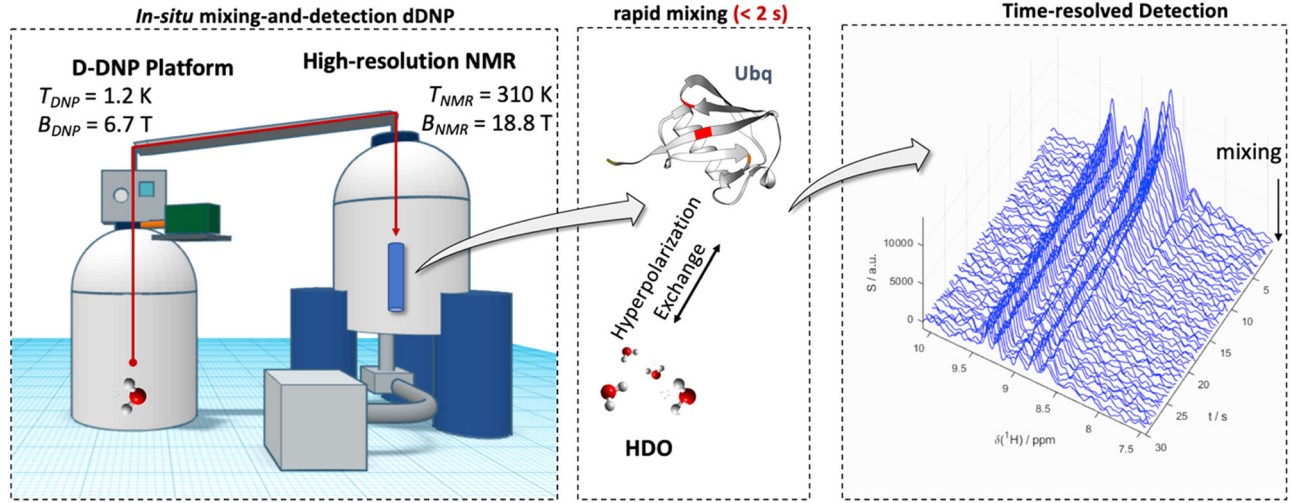

**Fig. 1 Experimental strategy for real-time hyperpolarized protein NMR.** Water is hyperpolarized at a temperature $T_{DNP}$ of 1.2 K and a magnetic field $B_{DNP}$ of 6.7 T before it is dissolved and mixed with a Ubq solution waiting at $T_{NMR}$ of 310 K in an NMR spectrometer. After mixing, hyperpolarization is introduced into the protein by chemical and magnetic exchange. Hence, a time series of NMR spectra can be recorded with superior signal intensities. The right panel shows how the signal intensities of the protein are boosted ([1]H NMR in blue) before the water polarization returns to thermal equilibrium.

[1]H-hyperpolarization into the latter, thereby selectively enhancing NMR signals of residues with favorable solvent interactions[24,26].

The resulting signal enhancement that can be achieved is exemplified in Fig. 1 (right) for the model protein Ubiquitin (Ubq) under near-physiological conditions, i.e., in aqueous solution at physiological salt concentrations, at pH 7.4 and 37 °C. Boosted signal amplitudes can be observed directly after mixing with hyperpolarized water. With time, the signal amplitudes decay since the spin hyperpolarization relaxes toward thermal equilibrium. Many resonances even drop below the detection threshold after the signal boost has decayed. The physiological buffer conditions accelerate amide proton exchange and thus do not yield good signal intensities in non-hyperpolarized spin systems.

(ii)  *Selecting a set of residues for detection*
The hyperpolarization exchange efficiency during step (3) of the *d*-DNP procedure depends on various factors[17,24,26]. For a given buffer and temperature, the most important factors are the delay $d_1$ between successive NMR detections as well as the structure of the target protein, which determines solvent exposure, as well as chemical exchange and NOE rates. We exploit this feature by fine-tuning $d_1$ such that only a subset of residues with strong solvent interaction becomes sufficiently hyperpolarized for single-scan NMR detection.
To explore and put this dependency to use, we employed water-selective nuclear Overhauser spectroscopy (WS-NOESY; Fig. 2a). Its three functional elements are: (a) A presaturation pulse-sequence that deletes all protein polarization, i.e., protein NMR signals, (b) a mixing period $\tau_m$ during which polarization from the hyperpolarized buffer can be transferred to the protein and (c) a 2D [1]H-[15]N (HSQC) sequence for detection. This design allows quantification of the transferred polarization from the solvent to the protein at residue resolution. The WS-NOESY approach to determine polarization transfer has been used in a series of recent publications by Kadeřávek et al.[24] and Hilty et al.[16,25]. They showed that proton exchange between water and protein transfers polarization to the latter depending on residue-specific exchange rates and exchange-relayed NOEs.

Here we add time-traces of polarization exchange to the current picture. In particular, we investigate the dependence on the mixing time $\tau_m$. In line with our earlier work[24], this confirmed that the transfer kinetics critically depends on the efficiency of solvent interaction of a residue. Figure 2b displays the signal [1]H$^N$ build-up due to the polarization transfer for our experimental conditions at the example of two residues, for slow (Q2, left panel) and fast (L8, right panel) transfer (Supplementary Fig. 1). The build-up curves show how the polarization is transferred from the solvent to the protein. The signal intensity for mixing times of 500 ms are indicated (corresponding to the interscan delay in *d*-DNP experiments, *vide infra*). For residue Q2 only little polarization is replenished within this time, while the opposite holds for L8[31].

(iii)  *Impact on hyperpolarization efficiency*
For a protein in a hyperpolarized buffer, the kinetics of polarization transfer determines the hyperpolarization efficiency. In a *d*-DNP experiment, the mixing time corresponds to the delay between successive detections during which protein polarization is replenished. This is confirmed in Fig. 2c, which shows residue-resolved signal intensities found in WS-NOESY at $\tau_m = 0.5$ s. Yellow bars indicate residues observed in *d*-DNP experiments for an interscan delay of 0.5 s (cf. Fig. 3). The evident correspondence between both experiments demonstrates that proton exchange and weak NOE can account for site-specific protein signal enhancement in hyperpolarized buffers[24]. We capitalize on this effect as it enables selective enhancement of a subset of residues of a globular protein in a *d*-DNP experiment by exploiting the elimination of residue signals with slow polarization-exchange kinetics at short mixing times.

(iv)  *Residue and temporally resolved real-time NMR of proteins*

Signal suppression at short mixing times can be used to achieve real-time monitoring of hyperpolarized Ubq at residue resolution. Choosing $\tau_m \approx 0.5$ s, the spectra become so sparse that only 13 signals can be observed in the WS-NOESY experiments. Thus, it is possible to deconvolute the signals of individual residues even in one-dimensional NMR. The correspondence between the signals observed in a WS-NOESY at a $\tau_m$ of 0.5 s and the observed

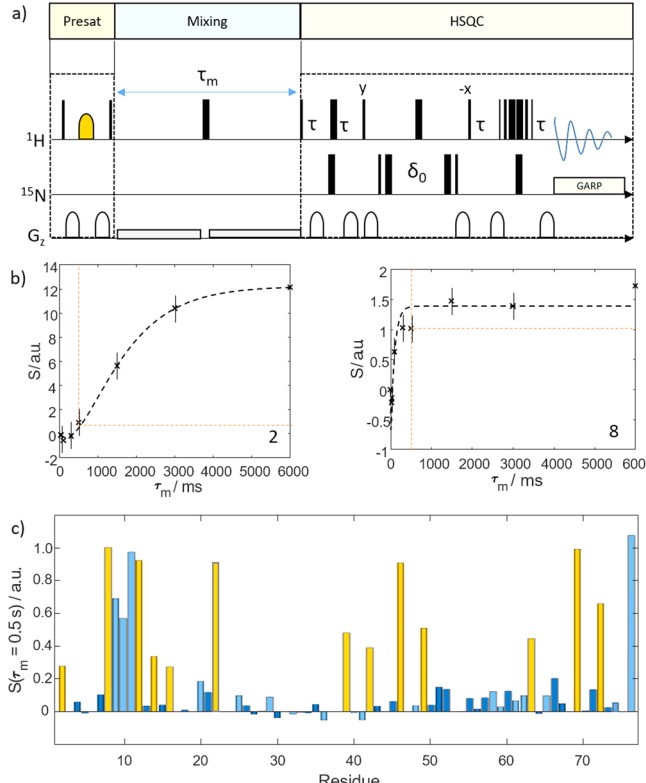

**Fig. 2 Water selective nuclear Overhauser spectroscopy. a** WS-NOESY pulse sequence used to measure the efficiency of the polarization transfer from water to the target protein. Thin black bars indicate 90˚ and thick bars 180˚ pulses. The yellow shape indicates a water-selective 180˚ pulse. Gray shapes indicate gradients. In the first segment, the protein polarization is deleted (yellow box). Next, during a mixing delay $\tau_m$, (blue box). polarization can be transferred from the water to the protein. Finally, the polarization is read out by means of a ¹H-¹⁵N Fast-HSQC spectrum (green box). Thus, by varying $\tau_m$, the kinetics of the polarization transfer can be assessed. The long weak gradients were used for radiation damping suppression. More details can be found in the Experimental section. **b** Experimental time traces (black dashed lines; signal intensity $S$ vs. mixing time $\tau_m$) for residues 2 and 8 showing signal build-up due to magnetic and chemical exchange with the water pool. The differences in build-up kinetics, which we put to use here, are evident. The signal intensity for a mixing time of 0.5 s is indicated by the dashed orange line. Data for all other residues are shown in the Supporting Information. The error bars correspond to the noise levels of the NMR spectra. **c** Residue-resolved signal intensities $S$ for a mixing time of 0.5 s. The yellow bars indicate the residues that where observed in the d-DNP experiments (see main text). The light blue bars indicate residues with resonance frequencies outside the excitation bandwidth in the d-DNP experiments.

signals in a ¹⁵N-edited ¹H 1D spectrum is shown in Fig. 3a. The 13 2D cross-peaks project onto ten discernible lines in the 1D spectrum. This allows the assignment of most resonances in the latter. Indeed, seven residues can unambiguously be identified (see Table 1). Due to this, the following experiment was possible:

Hyperpolarized water was first mixed with a Ubiquitin solution. After mixing, a series of ¹⁵N-edited ¹H spectra were recorded with an interscan delay of 0.5 s (according to $\tau_m$ in the sparse WS-NOESY). Figure 3b shows a selection of 1D spectra from this time series and corresponding fits of ten Lorentzians to the spectrum (Supplementary Figs. 2–3). Due to the spectral sparseness, the signals become deconvolutable, and it is possible to follow the hyperpolarization level of individual residues in real-time. The

determined residue-resolved signal enhancements are mapped onto the structure of Ubq in Fig. 3c. The corresponding time-dependent signal intensities are displayed in Fig. 3d. The figure shows how the signals are enhanced after mixing due to the exchange of hyperpolarization (at $t = 0$). With time the polarization decays toward thermal equilibrium.

The observed decay rates are functions of the water proton and ¹Hᴺ relaxation rates as well as of the proton exchange rate and detection pulse angles as shown by Szekely et al.[17]. Since the first two factors are residue dependent, varying apparent hyperpolarization decay rates can be observed for individual residues. Qualitatively, we observe that residues buried in the hydrophobic regions of the protein show faster relaxation due to less efficient replenishment of hyperpolarized protons, while residues in flexible regions show slower apparent decays (e.g., residue 14: 0.7 s⁻¹ vs. residue 47: 0.11 s⁻¹; cf. Table 1). Generally, the decay rates are low enough to detect boosted signals for 20 s after mixing the protein and the hyperpolarized buffer.

Besides, the buffer pH, temperature, and ionic strength can also influence proton exchange rates significantly[17,26]. These factors will, therefore, also impact the obtained signal enhancements. While higher pH and temperature typically lead to faster chemical exchange and more efficient transfer of hyperpolarization from the buffer, too rapid exchange can reduce the obtained signal intensities by leading to a loss of magnetization due to accelerated relaxation during pulse sequence evolution and detection. For the current study we chose physiological saline as a buffer to approach near-physiological conditions. The relationships between signal enhancement, type of residue, and contributions of NOE effects for Ubiquitin dissolved in a this buffer have been discussed in detail by Kadeřávek et al.[24]. It was shown that (under the experimental conditions used herein) an increase in exchange rate by a factor of five leads to a 10–15-fold increase in signal enhancement, as long as the rates remained below 10 s⁻¹.

Besides, it was found that solvent-relayed NOEs contribute to enhancing a small subset of residues (including D39, Q49, and K63) that feature protic side chain moieties, thereby boosting the signal intensity even further. This is also in line with the finding that residues with high enhancements here (see Table 1) feature such side chains. Yet, Kadeřávek et al.[24] also found that the major contribution to the signal enhancement is due to chemical proton exchange in fast exchanging residues localized in loop 1 and β sheet 2.

Note that this work[24] reported a single static hyperpolarized 2D spectrum at residue resolution. In contrast, herein we succeeded in recording (sub-second) time-resolved NMR data of a protein while neither giving up residue-resolution nor the signal-boost obtained from DNP. This combination enables us to resolve the state of various residues simultaneously at a 2 Hz sampling rate.

## Conclusions

Concluding, with the proposed technique, one can monitor hyperpolarized proteins at residue and high-temporal resolution to track individual diagnostic peaks, reminiscent of selective labeling strategies.

Various applications ranging from determining structural dynamics to drug discovery have significantly profited from such experimental strategies (see, e.g., references[32,33]). Here we aim to capitalize on a similar approach to sparsify protein spectra.

The proposed technique might not be applicable for intrinsically disordered proteins with narrow chemical shift dispersions in the proton dimension. Indeed, earlier work showed that for such proteins 2D or 3D detection is necessary to achieve residue-resolution, albeit lacking time resolution. However, ca. >70% of the proteome are well-folded, and, therefore, the presented technique is applicable to the largest share of target proteins. This approach might thus attract

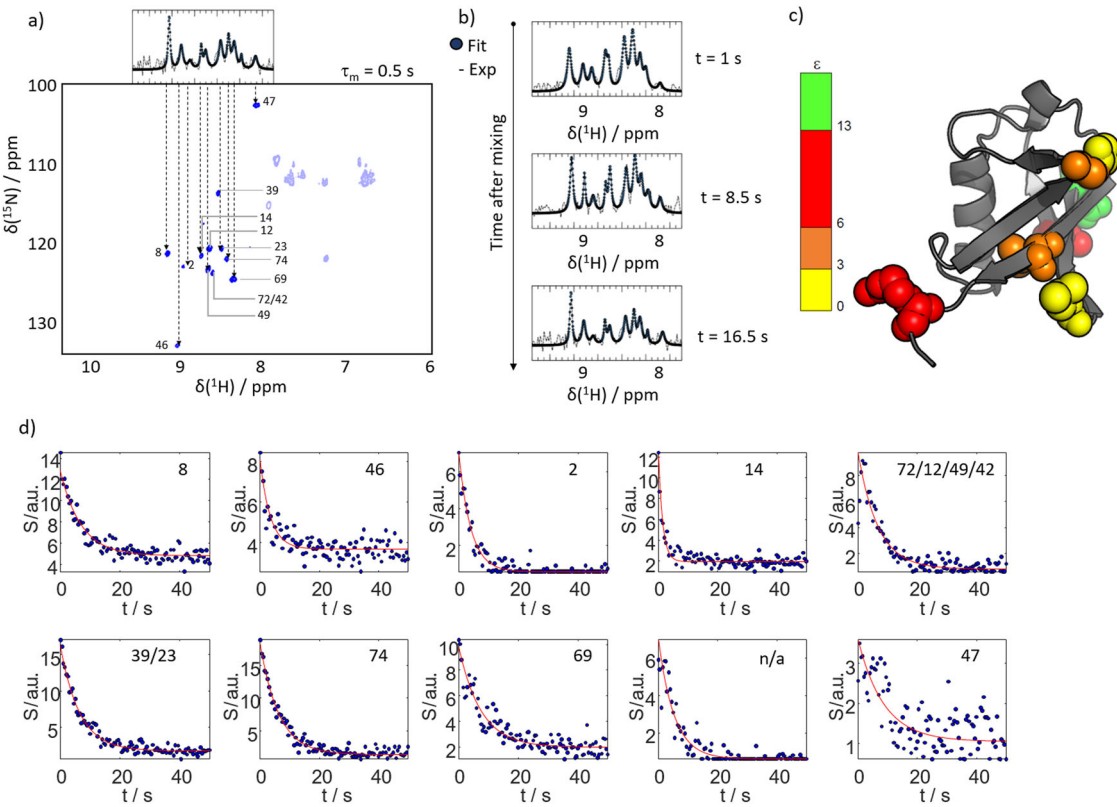

**Fig. 3 Real-time monitoring of hyperpolarization. a** With an interscan delay of 0.5 s only ten lines can be detected in hyperpolarized water as shown in the top insert (spectrum as black line, fit to 10 Lorentzians as blue dots). A water-selective NOESY spectra with a mixing time of $\tau_m = 0.5$ s confirms that only few residues are detectable. For most amino acids and polarization exchange and relaxation effects exactly cancel. Due the sparseness of the spectra most signals in the one-dimensional spectrum can be assigned. The transparent peaks fall out of the detected bandwidth in the d-DNP experiment due to the employed selective amide proton pulses. **b** $^{15}$N-edited proton 1D spectra of the ten observable H$^N$ lines (black) for different times after mixing with hyperpolarized water (using an interscan delay of 0.5 s; Spectrum as black line, fit to ten Lorentzians as blue dots). **c** Signal enhancements found for the observable residues mapped on the structure of Ubq. The color code indicates the observed signal enhancement, $\varepsilon$. **d** Time traces of signal intensities (blue dots) for residues the observable lines in the d-DNP experiments. The residue indices are indicated. Two lines could not unambiguously be assigned. Another line could not be assigned as no corresponding signal was found in the WS-NOESY (labelled 'n/a'). The red line indicates a fit to a monoexponential decay function with a rate $R_{1,app}$.

| Table 1 Enhancement factors and apparent decay rates of the ten fitted lines. | | | | | | | | | | |
|---|---|---|---|---|---|---|---|---|---|---|
| **Residue** | **L8** | **A46** | **Q2** | **T14** | **L71/T12/Q49/R42** | **D39/I23** | **R74** | **L69** | **n/a** | **G47** |
| $\varepsilon$ | 2.7 | 2.2 | >11.6 | 6.2 | 12.4 | 9.1 | 12.8 | 5.0 | >11.6 | 3.4 |
| $R_{1,app}/s^{-1}$ | 0.14 | 0.29 | 0.25 | 0.7 | 0.15 | 0.16 | 0.16 | 0.13 | 0.19 | 0.11 |

widespread interest as the technique is applicable to a large share of NMR studies.

We want to stress that the presented technique should not be considered as an alternative to established solution-state techniques but instead, as a complementary approach that can be put to use when protein real-time monitoring at residue-resolution is required. While we report a purely methodological advance herein, as an outlook, we anticipate use of this method in, e.g., time-resolved ligand-protein binding studies by mixing the target protein with a hyperpolarized solution of a binding partner. Indeed, most ligand interaction sites (e.g., binding pockets in drug design applications) or catalytically active sites remain incorporated in folded regions, where the presented approach is applicable. In such cases, the proton pool can boost signal intensities while the binding event proceeds. Indeed, real-time data can be obtained on a millisecond to seconds time scale, which is often impossible to obtain by NMR spectroscopy in thermal equilibrium.

## Methods

**Protein expression**. All protein samples were expressed and purified following the protocol published in reference[24]. In brief, $^{15}$N-enriched ubiquitin was expressed in E. coli BL-21-(DE3)-pLysS in an M9 medium containing 1 g/L $^{15}$N ammonium chloride as the only source of nitrogen. Ion exchange chromatography and gel filtration provided a pure protein sample. The protein was then concentrated using a 3.5 kDa cut-off Centricon to a final concentration of 1 mM in physiological saline at pH 7.4.

**Dissolution DNP**. 180 μL of a 15 mM TEMPOL solution in a 0.85:0.15 v/v mixture of H$_2$O and glycerol was vitrified in liquid helium. The glassy sample was positively hyperpolarized at 1.2 K in a magnetic field of 6.7 T by partial saturation of the EPR spectrum at 187.7 GHz (Bruker Biospin, ENS Paris). The microwave was modulated at 1 kHz with a saw-tooth function over a bandwidth of 100 MHz. Under these conditions, the hyperpolarization build-up rates were found to be $T_{build-up} = (2.1 \pm 0.1) \cdot 10^4$ s. Hence hyperpolarization build-up times of 3 h were used. Dissolution was achieved with a burst of 5 mL D$_2$O at 180 °C (453 K) under 1.05 MPa. The sample was subsequently propelled to a detection NMR spectrometer operating at 18.8 T (800 MHz for $^1$H) and 37 °C (310 K) within 1.5 s with He gas under 0.7 MPa through a "magnetic tunnel"[34] maintaining a constant magnetic

field of 0.9 T over a distance of approximately 4 m. The delay for mixing with the target protein solution (150 μL) and settling of turbulence prior to initiation of the NMR experiments was 3 s. After dissolution, the $H_2O:D_2O$ ratio was 0.03:1.

For detection, the pulse sequence shown in Supplementary Fig. 4 was used. For the $^1H$ channel, a selective 1000 μs long PC90 pulse was used for 90° excitation, and a 2000 μs long REBURP pulse was used for inversion. The carrier frequency was set to 10 ppm to avoid pulsing on the water resonance. Hence, only signals with chemical shifts >8 ppm were detected. Single scan detection was used, no phase cycling was applied. All data were apodized prior to Fourier transformation using a Guassian window function. For detecting the $^{15}N$-edited $^1H$-1D time series, a selective 1000 μs long PC90 pulse was used for 90° excitation, and a 2000 μs long REBURP pulse was used for inversion on the $^1H$ channel. The carrier frequency was set to 10 ppm to avoid pulsing on the water resonance. The rectangular shapes indicated 90° pulses. Hence, only signals with chemical shifts >8 ppm were detected. d2 was set to 0.00345 s, d1 to 0.5 s, and d0 to 0.00002780 s. d0 was not incremented. The FID was detected for 0.1 s during GARP decoupling. Note that similar experiments with a yet different detection scheme, i.e., a conventional SOFAST HMQC of BEST-HNCO led to the observation of similar sets of signals[24,29].

**Data analysis**. The time traces in Fig. 3 and Supplementary Figs. 1, and 2 were fitted using the MATLAB 'fit' function and the implemented least-square fitting routine using the least absolute residuals methods.

The spectral data in Supplementary Fig. 3 were fitted with Lorentzian functions using the MATLAB 'fitnlorezian' function[9,35]. We used least-square fitting as a statistical procedure to find the best fit by minimizing the sum of the residuals.

**NMR spectroscopy**. For the $^1H$-$^{15}N$ WS-NOESY experiments the pulse sequence of Fig. 2a was used.

The presaturation block selectively saturates all non-water protons of the system with a π/2 - z-gradient - π - z-gradient - π/2 sequence. For the water-selective π pulse, a 7.5 ms long Gaussian-shaped pulse was used with a carrier frequency centered at 4.7 ppm. The $^{15}N$ pulses were centered at 118.5 ppm. During the mixing delay, the 180° pulse compensates for relaxation effects, and the weak gradients suppress radiation damping. The HSQC block is a conventional gradient-selective pulse sequence using the 3-9-19 Watergate for solvent suppression. All spectra were recorded on a Bruker NEO spectrometer operating at a proton Larmor frequency of 600 MHz. The spectrometer was equipped with a prodigy TCI probe. The π/2 pulse length was 11.1 μs for the $^1H$ channel and 37 μs for the $^{15}N$ channel. The temperature was set to 298 K. 10% $D_2O$ was used as lock solvent in an otherwise physiological saline at pH 7.4. In total, 128 complex increments were detected in the indirect dimension. All data were apodized with a 60° shifted sine-bell functions and zero-filled prior to Fourier transformation to twice the FID length. Subsequently, data were baseline corrected along the directly detected dimension using NMR Pipe[36].

All experiments were repeated three times in independent replica runs. The results were always similar.

## Data availability

All data are available under the open access license Creative Commons Attribution 4.0 International, under the DOI 10.5281/zenodo.5547224. The data contains files in MATLAB and Bruker TopSpin 4 format.

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

## Acknowledgements

The project leading to this application received funding from the European Research Council (ERC) under the European Union's Horizon 2020 research and innovation programme (grant agreement 801936). This project was supported by the Austrian FWF (stand-alone grant no. P-33338). The authors thank Dr. Pavel Kadeřávek and Dr. Gregory L. Olsen for helpful discussions. Furthermore, the authors thank Prof. Geoffrey Bodenhausen for his support from the beginning of this research programme on.

## Author contributions

M.N. and D.K. performed experiments. D.K. wrote the paper with the help of M.N.

## Competing interests

The authors declare no competing interests.
