## [Peer Review File · Communications Chemistry]

REVIEWERS' COMMENTS:

Reviewer #1 (Remarks to the Author):

The Kurzbach group has reported NMR observations with high temporal resolution by hyperpolarization of proteins. They have previously reported high sensitivity for folded proteins using dissolution DNP, and have now extended this to physiologically relevant conditions in a buffer, enabling fast observations in the order of seconds. Although the application of dissolution-DNPs to solution NMR is very important, it has not yet been widely studied, but this report is expected to expand the application to a variety of biomolecules.

Although I did not find any particular problem with the content of the paper, it would be desirable to have a more detailed explanation of the differences between this paper and the past work of the authors, especially the Chem. Eur. J. paper. It would be better to add a quantitative discussion of how the use of buffer affected the polarization enhancement obtained.

Reviewer #2 (Remarks to the Author):

The work by M. Negroni and D. Kurzbach concerns the use of hyperpolarized buffers (water in this case), via dDNP, to enhance sensitivity in 2D NMR spectroscopy of proteins at residual resolution. Using hyperpolarized water to enhance 2D NMR spectroscopy of proteins has gained increasing interest during the last 5 years. The authors' main claim is to add a time aspect to this methodology. Indeed, by playing with the mixing time, during which water protons exchange polarization with the nuclei born on the protein, they manage to acquire NMR spectra that are sparse enough to monitor in real time the behavior of specific amino acids (residue resolution), with both 2D and 1D NMR spectroscopy at physiological conditions. This technique has potential use as a probe for e.g. time-resolved ligand-protein study.

Communication Chemistry is a journal with a broad scope and a broad audience. As an expert in hyperpolarization, but not in protein NMR I found somewhat difficult to clearly understand the novelty of the method compared to previous works such as ref 16, where a kinetic study of proteins using 2D NMR and hyperpolarized water was performed. I am not complaining about the quality of the work, but the authors must improve the introduction and make the scope more clear, should this work be accepted in Communication Chemistry. Differently, I suggest some more specific journals as JMR. Therefore, I ask the author to add a couple of paragraphs to better explain where previous works end and how and to which extent this work provides something new. Moreover, in order to proof the originality of the work and strengthen its value, I suggest to perform experiments concerning the "time resolved ligand-protein binding" application they mention in the conclusion. This will for sure strengthen the value of the paper, make conclusions convincing, and have higher impact in the field.

More specific comments follow below:

Main text

page 1, 2nd column: replace "longer" with "long", otherwise provide the term of comparison.

page 1, 2nd column: replace "however" with "therefore".

page 2, column 2: please expand on why "dDNP has to give up the major advantage...". This is a key point of the paper. Just stating it it's not enough. Do you refer to direct dDNP of biomolecules, or you already consider the use of hyperpolarize water?

page 3, column 2: please complete as follows "achieved by slightly off-resonance microwave irradiation of the radical ESR line".

page 4, column 1, line 5: "where it mixes" rather than "where it mixed"

page 5, column 1, line 38: add comma after "dependent"

Experimental

page 6, column 1, line 2: "v/v" instead of "v/v/v"

page 6, column 1, line 6: was microwave frequency modulation used? if so specify parameters

page 6, column 1, line 8: provide build up time instead of buildup rate, since you mention a polarization time of 3h right after.

page 6, column 1, line 21: "0.03:1" instead of "1:0.003"

In the experimental part there is mentioned nowhere how many times the experiments were repeat and the kind of statistical analysis performed.

Figure Legends

Figure 2, line 7: after "thus" remove "the"

Figure 3, line 2: "inset" instead of "insert"

Reviewer #3 (Remarks to the Author):

Manuscript: COMMSCHEM-21-0263

Title: Residue-resolved monitoring of protein hyperpolarization at sub-second time resolution

Authors: Mattia Negroni, Dennis Kurzbach

Dissolution DNP is one of the most powerful techniques for the signal enhancement of NMR spectroscopy. In its current version it consists usually of two steps: in the first step, the frozen target system is hyperpolarized via microwave irradiation of stable radicals in a polarization magnet at cryogenic temperatures. In the second step the hyperpolarized target is rapidly thawed and inserted into the detection magnet. Two major drawbacks of this techniques are the loss of resolution and non-physiological conditions.

The current manuscript addresses both these problems by hyperpolarizing not the target system but the solvent (water) which is rapidly mixed with the target molecule after hyperpolarization. The target molecule is then hyperpolarized by polarization transfer from the hyperpolarized water.

This is a very well-written manuscript which is clearly of high interest to the NMR and the biochemical community. Recommendation: publish as is

Minor comments: the authors might check the graphic quality of the spectra in the supporting information, which look rather blurry from the compression in the pdf obtained for reviewing.

Reviewer #1 (Remarks to the Author):

The Kurzbach group has reported NMR observations with high temporal resolution by hyperpolarization of proteins. They have previously reported high sensitivity for folded proteins using dissolution DNP, and have now extended this to physiologically relevant conditions in a buffer, enabling fast observations in the order of seconds. Although the application of dissolution-DNPs to solution NMR is very important, it has not yet been widely studied, but this report is expected to expand the application to a variety of biomolecules.

We thank the referee for his positive assessment of our work.

Although I did not find any particular problem with the content of the paper, it would be desirable to have a more detailed explanation of the differences between this paper and the past work of the authors, especially the Chem. Eur. J. paper.

We have expanded the discussion about the differences of the previous paper published in Chem. Eur. J. In fact, the earlier work reported a static spectrum that achieved residue-resolved data (as in conventional NMR), but at better signal amplitudes. In stark contrast, we succeeded in recording (sub-second) time-resolved NMR data of a protein while neither giving up residue-resolution nor the signal-boost obtained from DNP. This combination has (to the best of our knowledge) not been achieved by any other technique before and enables us to resolve the state of various residues simultaneously at a 2 Hz sampling rate.

It would be better to add a quantitative discussion of how the use of buffer affected the polarization enhancement obtained.

We thank the referee for this comment. We now realized that we missed referring to the recent body of work and to comment on buffer effects on hyperpolarization exchange rates. Following the referee's suggestion, we have amended the manuscript with a paragraph on the matter. It reads:

"Besides, the buffer pH, temperature, and ionic strength can also influence proton exchange rates significantly.^{17,26} These factors will, therefore, also impact the obtained signal enhancements. While higher pH and temperature typically lead to faster chemical exchange and more efficient transfer of hyperpolarization from the buffer, too rapid exchange can reduce the obtained signal intensities by leading to a loss of magnetization due to accelerated relaxation during pulse sequence evolution and detection. For the current study we chose physiological saline as a buffer to approach near-physiological conditions. The relationships between signal enhancement, type of residue, and contributions of NOE effects for Ubiquitin dissolved in a this buffer have been discussed in detail by Kadeřávek et al. (see reference ²⁴). It was shown that (under the experimental conditions used herein) an increase in exchange rate by a factor of 5 leads to a 10 to 15-fold increase in signal enhancement, as long as the rates remained below 10 s⁻¹."

However, the dependence of the enhancement depends on a multitude of factors, including the pulse sequence used, the studied protein, and the spectrometer frequency. Hence, we would like to refrain from making any more general statements about this dependency.

Reviewer #2 (Remarks to the Author):

The work by M. Negroni and D. Kurzbach concerns the use of hyperpolarized buffers (water in this case), via dDNP, to enhance sensitivity in 2D NMR spectroscopy of proteins at residual resolution. Using hyperpolarized water to enhance 2D NMR spectroscopy of proteins has gained increasing interest during the last 5 years. The authors' main claim is to add a time aspect to this methodology. Indeed, by playing with the mixing time, during which water protons exchange polarization with the nuclei born on the protein, they manage to acquire NMR spectra that are sparse enough to monitor in real time the behavior of specific amino acids (residue resolution), with both 2D and 1D NMR spectroscopy at physiological conditions. This technique has potential use as a probe for e.g. time-resolved ligand-protein study.

Communication Chemistry is a journal with a broad scope and a broad audience. As an expert in hyperpolarization, but not in protein NMR I found somewhat difficult to clearly understand the novelty of the method compared to previous works such as ref 16, where a kinetic study of proteins using 2D NMR and hyperpolarized water was performed.

Reference 16 by Hilty and co-workers reports protein data at low spectral resolution. The novelty of our work lies in the combination of time-resolved dissolution DNP with residue resolution. Residue resolution is at the heart of biomolecular NMR, and amending real-time dDNP with this possibility opens entirely new avenues for biomolecular and structural biology as protein structural dynamics can be assessed. This possibility was not provided by reference 16. It should be stressed, though, that reference 16 focuses on determining proton exchange rates and does not aim to resolve individual residues. We have now included a sentence explaining this circumstance (see also the following comment/response).

I am not complaining about the quality of the work, but the authors must improve the introduction and make the scope more clear, should this work be accepted in Communication Chemistry. Differently, I suggest some more specific journals as JMR. Therefore, I ask the author to add a couple of paragraphs to better explain where previous works end and how and to which extent this work provides something new.

We did as the referee suggested and described the scope of the manuscript in the framework of the established methodology in the revised introduction. In particular, we explain how the current paper forwards the field. The new paragraph reads:

"The use of hyperpolarized water to boost signal intensities in multidimensional NMR spectra of protein was first suggested by Frydman and co-workers in 2014.²³ In 2017, the approach was shown to be capable of yielding residue-resolved spectra of proteins and intrinsically disordered proteins^{15, 17}, yet, only a single static spectrum could be detected. At the same time, it was shown by Hilty and co-workers that signal-boosted and time-resolved spectra could be obtained, albeit low-resolution.¹⁶ Later, the hyperpolarized water technique was expanded to folded targets²⁴. Since then, several applications of the technique have been developed, including the characterization of folding intermediates,²⁵ exchange processes,²⁶ and membrane interactions²⁷. Herein, the use of hyperpolarized water to boost biomolecular NMR spectra is expanded by enabling both time- and residue-resolution in a single experiment. Similar approaches have also been successfully implemented for RNAs²¹."

Moreover, in order to proof the originality of the work and strengthen its value, I suggest to perform experiments concerning the "time resolved ligand-protein binding" application they mention in the conclusion. This will for sure strengthen the value of the paper, make conclusions convincing, and have higher impact in the field.

We consider the mentioned ligand binding experiments as an outlook highlighting possible future applications for the reader. As stated in our response above, the key point of this manuscript is a methodological one, namely, to combine DDNP with residue-resolved data. Without this development, the combination of DNP with bioNMR will remain of very limited scope. We add another phrase to the manuscript to make this point clear.

More specific comments follow below:

Main text

page 1, 2nd column: replace "longer" with "long", otherwise provide the term of comparison.

We followed the referee's suggestions.

page 1, 2nd column: replace "however" with "therefore".

We followed the referee's suggestions.

page 2, column 2: please expand on why "dDNP has to give up the major advantage...". This is a key point of the paper. Just stating it it's not enough. Do you refer to direct dDNP of biomolecules, or you already consider the use of hyperpolarize water?

We were referring to the use of hyperpolarized water and its applications to biomolecules. So far, only two types of experiments have been possible: 1. Detection at low spectral resolution, but high temporal resolution; 2. the other way around, at low temporal resolution but high spectral resolution. This compromise was necessary since for a 2D or 3D spectrum several t_1 -increments need to be recorded to construct the indirect dimensions slowing down acquisition of a full spectrum. Hence, when recording residue-resolved protein spectra, the time resolution, which is considered a primary advantage of dDNP, had to be given up. Alternatively, one can detect a high-resolution multidimensional spectrum, albeit acquisition times of ca. 20-30 s. To make this point clear now, we have added another paragraph to the manuscript. It reads:

"In this regard, dDNP provides significantly improved signal intensities outdating the need for signal averaging. Spectra can be recorded rapidly, *i.e.*, within milliseconds to seconds, yet only when relying on one-dimensional detection. Thus, acquiring NMR spectra in real-time NMR becomes possible, yet residue-resolved 2D or 3D spectra of proteins can typically not be recorded. *Vice-versa*, hyperpolarized spins have a limited lifetime (typically seconds to minutes), such that detection is possible only within a short time window. Hence, when recording fast 2D or 3D protein spectra by

acquiring several signals to construct the indirectly detected dimensions, measuring a single spectrum takes too long for time-resolved measurement series.

Hence, herein, we present a novel strategy that aims to overcome this predicament by using hyperpolarized water to enhance nuclear magnetic resonance proton signals in proteins."

page 3, column 2: please complete as follows "achieved by slightly off-resonance microwave irradiation of the radical ESR line".

We amended the manuscript following the referee's suggestion.

page 4, column 1, line 5: "where it mixes" rather than "where it mixed"

We followed the referee's suggestions.

page 5, column 1, line 38: add comma after "dependent"

We followed the referee's suggestions.

Experimental

page 6, column 1, line 2: "v/v" instead of "v/v/v"

We amended the manuscript following the referee's suggestion.

page 6, column 1, line 6: was microwave frequency modulation used? if so specify parameters

We added this information to the manuscript. Microwave frequency modulation was used (100 MHz saw-tooth modulation).

page 6, column 1, line 8: provide build up time instead of buildup rate, since you mention a polarization time of 3h right after.

We amended the manuscript following the referee's suggestion.

page 6, column 1, line 21: "0.03:1" instead of "1:0.003"

We amended the manuscript following the referee's suggestion.

In the experimental part there is mentioned nowhere how many times the experiments were repeat and the kind of statistical analysis performed.

We added this information to the Experimental section.

Figure Legends

Figure 2, line 7: after "thus" remove "the"

Figure 3, line 2: "inset" instead of "insert"

We corrected this.

Reviewer #3 (Remarks to the Author):

Manuscript: COMMSCHEM-21-0263

Title: Residue-resolved monitoring of protein hyperpolarization at sub-second time resolution

Authors: Mattia Negroni, Dennis Kurzbach

Dissolution DNP is one of the most powerful techniques for the signal enhancement of NMR spectroscopy. In its current version it consists usually of two steps: in the first step, the frozen target system is hyperpolarized via microwave irradiation of stable radicals in a polarization magnet at cryogenic temperatures. In the second step the hyperpolarized target is rapidly thawed and inserted into the detection magnet. Two major drawbacks of this techniques are the loss of resolution and non-physiological conditions.

The current manuscript addresses both these problems by hyperpolarizing not the target system but the solvent (water) which is rapidly mixed with the target molecule after hyperpolarization. The target molecule is then hyperpolarized by polarization transfer from the hyperpolarized water.

This is a very well-written manuscript which is clearly of high interest to the NMR and the biochemical community. Recommendation: publish as is

We thank the referee for his motivating words.

Minor comments: the authors might check the graphic quality of the spectra in the supporting information, which look rather blurry from the compression in the pdf obtained for reviewing.

We will provide high-resolution figures when the manuscript is accepted for publication.